# Shedding light on development: Leveraging the new nightlights data to measure economic progress

Prachi Jhamb[1][�]*, Susana Ferreira[1][�], Patrick Stephens[2][‡], Mekala Sundaram[3][‡], Jonathan Wilson[4][‡]

1 Department of Applied Economics, University of Georgia, Athens, Georgia, United States of America,
2 Integrative Biology, Oklahoma State University, Stillwater, Oklahoma, United States of America,
3 Department of Infectious Diseases, University of Georgia, Athens, Georgia, United States of America,
4 Department of Pathology, University of Georgia, Athens, Georgia, United States of America

☩ These authors contributed equally to this work.
‡ PS, MS and JW also contributed equally to this work.
* pj40553@uga.edu

**Data Availability Statement:** Data for the DHS wealth index cannot be shared publicly due to restrictions outlined by the Demographic and

## Abstract

Nightlights (NTL) have been widely used as a proxy for economic activity, despite known limitations in accuracy and comparability, particularly with outdated Defense Meteorological Satellite Program (DMSP) data. The emergence of newer and more precise Visible Infrared Imaging Radiometer Suite (VIIRS) data offers potential, yet challenges persist due to temporal and spatial disparities between the two datasets. Addressing this, we employ a novel harmonized NTL dataset (VIIRS + DMSP), which provides the longest and most consistent database available to date. We evaluate the association between newly available harmonized NTL data and various indicators of economic activity at the subnational level across 34 countries in sub-Saharan Africa from 2004 to 2019. Specifically, we analyze the accuracy of the new NTL data in predicting socio-economic outcomes obtained from two sources: 1) nationally representative surveys, i.e., the household Wealth Index published by Demographic and Health Surveys, and 2) indicators derived from administrative records such as the gridded Human Development Index and Gross Domestic Product per capita. Our findings suggest that even after controlling for population density, the harmonized NTL remain a strong predictor of the wealth index. However, while urban areas show a notable association between harmonized NTL and the wealth index, this relationship is less pronounced in rural areas. Furthermore, we observe that NTL can also significantly explain variations in both GDP per capita and HDI at subnational levels.

## Introduction

National-level indicators of economic and social progress may provide a bird's eye view of a country's performance, but they often mask the heterogeneity that exists within countries. Subnational variation in resource endowments and development outcomes can have a

Health Survey (DHS) Program. Access to DHS data requires researchers to register and agree to the terms of use, which prohibit public redistribution. These data are available directly from the DHS Program (https://dhsprogram.com/data/) upon researcher registration and approval. Data on other variables used in this study, including Nightlights, Population Density, GDP per capita, and the Human Development Index, are publicly available and have been compiled by the authors as a cleaned dataset. This compiled dataset is available using the following Figshare repository URL: https://doi.org/10.6084/m9.figshare.28095584.v1.

**Funding:** This work was supported by National Institutes of Health, NIH R01AI156866 'Spillover of Ebola and other filoviruses at ecological boundaries' (Patrick Stephens lead investigator)." https://www.nih.gov/ The funder played no role in the study design, data collection or analysis, decision to publish, or preparation of the manuscript.

**Competing interests:** NO authors have competing interests.

significant impact on people's well-being, but their analysis is frequently overlooked due to the lack of reliable disaggregated administrative data [1].

The need for subnational data in developing countries is particularly critical in the context of infectious disease spillover and climate change. The spatial heterogeneity in socio-economic conditions, disease transmission patterns, and healthcare infrastructure within a country can have a significant impact on the effectiveness of public health response measures to control disease outbreaks [2, 3]. To advance knowledge of disease transmission and control, spatially referenced demographic information, including distinctions based on cohorts and gender, is essential [4]. Unfortunately, such data are frequently inaccessible and available only in countries that undertake comprehensive census surveys [5]. Similarly, in the context of climate change, whose impacts can be highly localized and vary across regions within the same country due to factors such as topography, climate variability, or land use, subnational data can help in identifying areas that are particularly vulnerable to climate impacts and inform resilience strategies [6]. Without subnational data, researchers in the past have relied on national level development indicators which is problematic since development indicators can conceal significant intra-national variation, particularly in large countries [7].

The average interval between nationally representative economic surveys in half African nations is above 6.5 years, compared to a sub-annual frequency in most wealthy countries [8]. Household surveys, such as those from the Demographic and Health Surveys (DHS) program have limited repeated observations of the same location (and even the same country) over time [9, 10]. It is estimated that a given African household would appear in a household survey once in 1,000 years, making it difficult to measure changes in well-being over time at a local level [11]. For this reason, researchers have turned to alternative and non-traditional sources of data such as nightlights (NTL), daytime satellite imagery or mobile phone call detail records [12–15].

NTL have been used as a proxy for economic activity in various studies based on the assumption that the amount of light at night is closely tied to the level of economic activity [16–19]. The reliance on NTL intensity extends beyond measuring economic activity, with researchers employing it in various applications such as studying inequality and assessing the accuracy of official statistics in different political contexts [20]. However, concerns about the reliability of NTL data have emerged, particularly due to the extensive use of outdated and inaccurate data from the now-discontinued Defense Meteorological Satellite Program (DMSP), which ceased production in 2013 [21].

DMSP data suffer from several limitations, including blurred images, geo-location errors, and top-coding, which result in misattributed light sources and an inability to distinguish between areas of low and high light intensity. These issues undermine the accuracy of analyses, as DMSP data often aggregate light intensities, masking important details [22]. This has highlighted the need for a transition to newer and more precise data sources, such as the Visible Infrared Imaging Radiometer Suite (VIIRS).

VIIRS data, available since 2012, offers substantial improvements over DMSP, including 45 times higher spatial resolution and the elimination of blurring and geo-location errors, making it a more reliable alternative for economic analyses [21, 23]. Despite these advantages, the adoption of VIIRS in the economics literature has been limited. Two primary factors contribute to this: Firstly, DMSP data still offers a longer time series spanning from 1992 to 2013, which is advantageous for time series analysis. Secondly, comparing DMSP and VIIRS data directly presents challenges due to differences in their temporal coverage and spatial resolutions [24]. While DMSP data is available annually, ending in 2013, VIIRS offers annual data only for 2015 and 2016. Monthly VIIRS data is available from 2012 onwards, but comparing monthly VIIRS data with annual DMSP data introduces potential inaccuracies due to differing data processing methodologies [24].

To address these challenges, this paper utilizes a novel harmonized NTL dataset developed to bridge the gap between DMSP and VIIRS datasets [25]. This dataset creates a continuous and consistent NTL time series from 1992 to 2021 with high spatial resolution (see S1 Appendix), enabling direct comparisons across global regions.

Our first contribution lies in testing the accuracy of the harmonized NTL dataset in measuring economic activity at a small spatial scale for countries in sub-Saharan Africa. Through this, we aim to provide insights into the effectiveness of the harmonized NTL dataset for economic analyses at fine spatial resolutions in developing regions. Previously, researchers tested the dataset's accuracy for predicting regional GDP in 2012 for Colombia [26]. We build on this work by focusing on subnational socio-economic activity in sub-Saharan Africa. We rely on household surveys conducted by the DHS as our primary source of information on subnational socio-economic activity. These surveys assess living conditions and household possessions and are available for multiple years and countries. It is worth noting that most previous studies evaluating the accuracy of satellite data compare them against the wealth index derived from DHS data [11, 12, 27, 28].

Our second contribution is that we control for population density. Criticisms of using NTL as a proxy for economic activity highlight the likelihood of high correlation between the density of light and population density [9]. However, few studies so far have evaluated the extent to which NTL vary independently of population density.

Our third contribution is an exploration into whether NTL can also serve as a proxy for human development outcomes, an area that has been largely underexplored in existing literature. While there are notable exceptions, such as [28, 29], these studies did not utilize the new harmonized NTL dataset. Moreover, their analysis of HDI relied on data constructed using indicators from the DHS itself, such as the wealth index, education, and health. In contrast, our analysis employs alternative datasets, which provide annual gridded data for HDI and GDP per capita, derived entirely from macroeconomic indicators rather than satellite imagery or household surveys. Specifically, we utilize datasets that provide annual gridded data for GDP per capita and the Human Development Index (HDI) (encompassing health, education, and standard of living) [7]. Importantly, our analysis benefits from the fact that the subnational indicators of economic activity, namely HDI and GDP per capita, are derived from macroeconomic indicators and are not computed using satellite imagery or household surveys.

The efficacy of NTL as a proxy for economic activity, particularly in rural areas, has been a subject of debate in the literature. While some studies found NTL to be unreliable proxies for GDP in rural settings in Indonesia [30], others argue for their reliability in rural areas of Colombia [26]. Finally, our study contributes to this ongoing debate by investigating the heterogeneity in the variation captured by the new harmonized NTL separately in urban versus rural areas and examining the usefulness of NTL as a proxy for economic activities at subnational levels in rural areas.

By quantifying the degree to which the newly available harmonized NTL data correlate with several indicators of economic activity at the subnational level (i.e. household wealth index derived from the DHS, and gridded HDI and GDP per capita) across 34 countries in sub-Saharan Africa, we aim to provide insights into the effectiveness of NTL as a proxy for economic activity. We also consider the degree to which NTL captures additional information beyond simple population density. Table 1 summarizes the datasets utilized in this study, along with their sources and temporal coverage.

Our focus on sub-Saharan Africa is motivated by its unique challenges in obtaining accurate subnational socioeconomic data and its inherent significance as a region experiencing rapid population growth and ongoing economic changes [33], along with a high risk of emerging human diseases [34].

**Table 1. Data sources and coverage.**

| Variable | Source | Spatial Coverage and Resolution | Temporal Coverage |
|---|---|---|---|
| Household wealth index | [31] | Over 90 countries, GPS coordinates, 10 km | Since 1980s; patchy |
| Nighttime lights data | [25] | Global, 1 km | 1992 to 2021; annual |
| Population density | [32] | Global, 5 km | 2000 to 2020; quinquennial |
| Human development index | [7] | 39 countries, 10 km | 1990 to 2015; annual |
| Gross domestic product per capita | [7] | 82 countries, 10 km | 1990 to 2015; annual |

## Materials and methods

### Data

**Demographic and Health Surveys (DHS) wealth index.** The primary dependent variable in our study is derived from the DHS, which are a series of nationally representative and standardized household surveys conducted in low-income countries in Africa and other regions since the 1980s to monitor and evaluate population, health, and nutrition programs. Our sample includes DHS surveys that have geo-coordinates to match them to NTL data. For a comprehensive list of the countries included in our sample, along with the corresponding years of data collection, please refer to S1 Table.

The DHS data are repeated cross-sections rather than a panel survey since a new sample of clusters is drawn for each round for the same country. The geographic information is available at the level of survey clusters or Primary Sampling Units (PSUs) rather than at the level of households. The clusters are categorized into urban and rural groups and the cluster location is reported with latitude/longitude coordinates [9]. Most cluster locations are measured using GPS while only some are measured by information from gazetteers [35]. However, to ensure anonymity, cluster location points are randomly displaced by up to 2 kms for urban clusters and by 5 kms for rural clusters, with less than 1 percent of rural clusters displaced by up to 10 km [28].

Our primary variable of interest is the DHS wealth index, which is derived from data on households' ownership of a selected set of assets such as a phone, radio, car, TV, or motorbike; dwelling characteristics like the number of rooms occupied in a home, flooring material, access to electricity, type of drinking water source, as well as other characteristics related to the wealth status. Specifically, we use the Household Recode (HR) survey data for each country, which contains all household attributes, alongside the corresponding GPS dataset for the same year and phase (for details on construction of DHS wealth index, see [31]).

The DHS wealth index is the most widely used variable to capture poverty in studies analyzing predictors of economic well-being [11, 27]. This is because, given the limited availability of comprehensive socioeconomic indicators with high spatial resolution across a wide range of developing countries, the DHS wealth index is considered the most suitable option [35]. However, due to reasons listed in the previous section, its coverage is patchy. S1 Table shows that in our sample of 34 countries between 2004 and 2019, Rwanda was surveyed in most years (5), followed by Ethiopia (4), with most countries surveyed only once.

The household-level wealth index factor score is calculated by analyzing the asset ownership of each individual through principal component analysis. A categorical household wealth index variable ranging from 1 to 5 is derived from the household-level wealth factor score where 1 represents the lowest asset levels or "poorest" households and 5 represents the highest asset levels or "richest" households. However, as stated above, the DHS doesn't provide household locations but rather the average location of a group of households which is referred to as

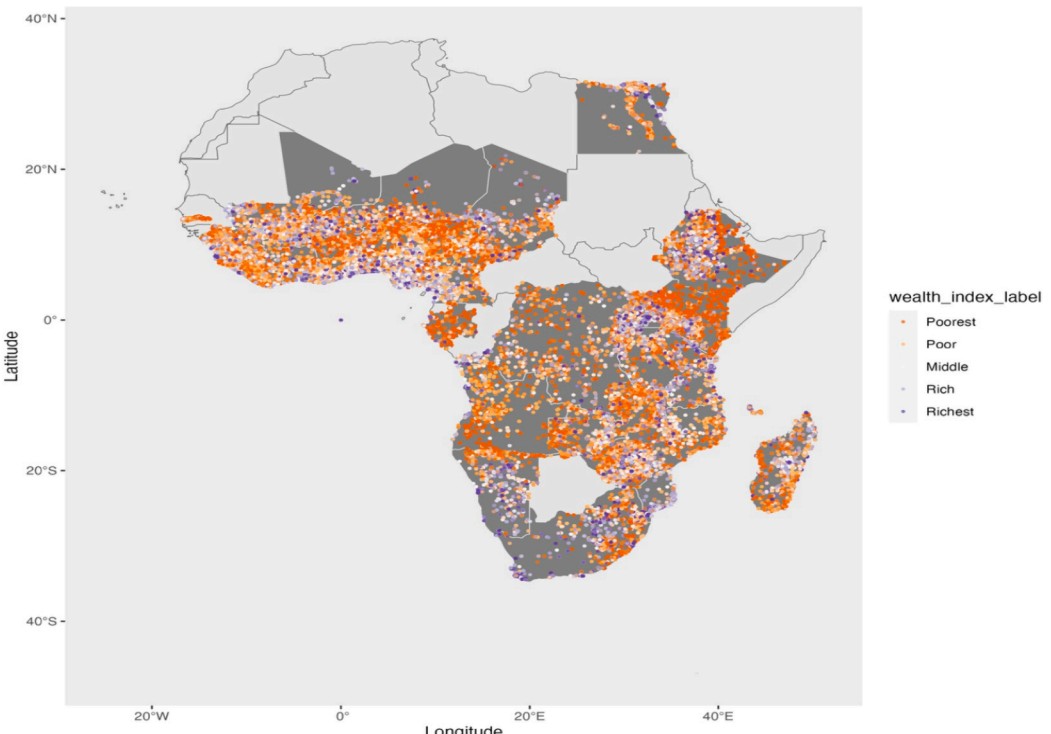

**Fig 1. Geographic distribution of DHS survey clusters in Africa.** Base map data from the spData package in R, derived from Natural Earth (public domain). Household Wealth Index data is derived from DHS. The figure depicts geographical distribution of DHS clusters in the sample, with points colored by household wealth category (Poorest to Richest). Countries shaded in gray represent those included in the study. The figure was created by the author.

a household cluster. Therefore, we extract the wealth index across all surveys in our sample and create average wealth index across all households within each cluster.

Fig 1 displays the geographical distribution of DHS clusters included in our sample. Each point on the map represents a cluster, and its color indicates the wealth category of the households surveyed. The countries shaded in gray represent those included in our sample. It is important to note that the DHS wealth index is a relative measure of wealth. It is constructed using country-specific methodologies, which limits its applicability for cross-country comparisons. Therefore, interpretations of the wealth index should be confined to within-country comparisons.

**Nightlights data.** We use a harmonized NTL dataset [25], which allows us to expand the time-period for our study from 2004 to 2019. The dataset is available yearly, downloadable as rasters at a spatial resolution of 30 arc-seconds (~1 km). Unlike the raw data from the two primary sources—DMSP (1992–2013) and VIIRS (2013–2019)—which are not directly comparable due to limited temporal overlap and differing spatial resolutions, this harmonized dataset bridges these gaps by simulating DMSP-like data from VIIRS. For details on the harmonization process and data sources, please refer to S1 Appendix.

By using this dataset, we not only extend the temporal coverage but also enhance the sensitivity of our nighttime light indicator. Their approach, particularly the simulation of DMSP-like data using VIIRS, improves the capture of medium-to-low light emissions, making it valuable for investigating local development impacts and economic output in Africa [36].

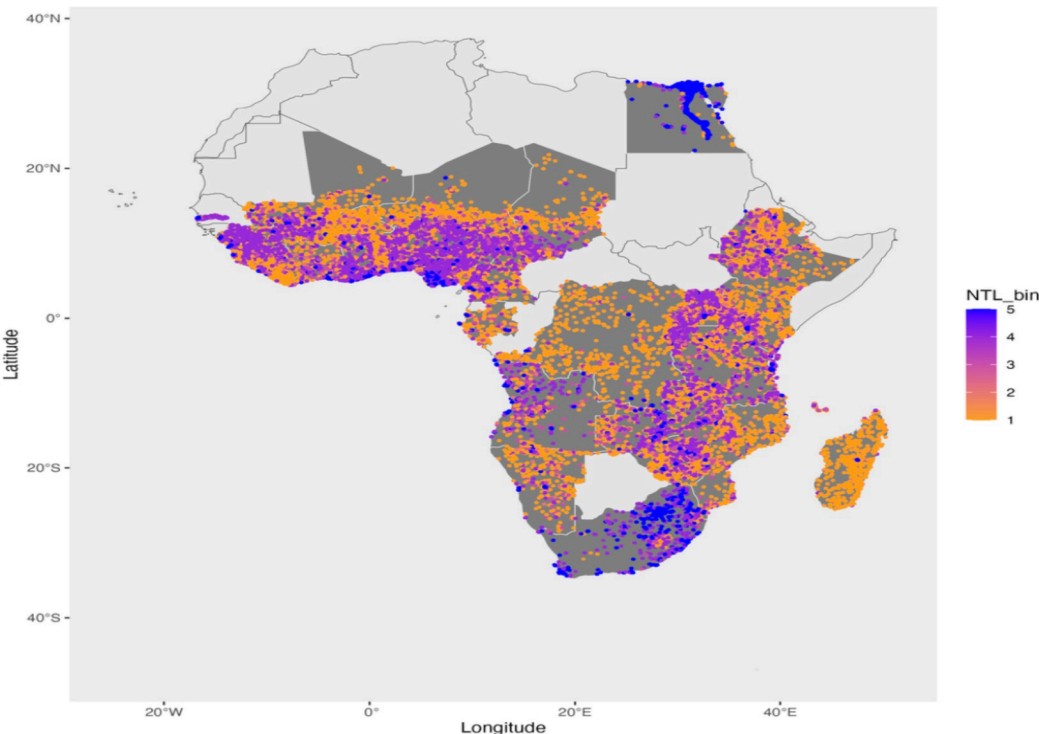

**Fig 2. Geolocated DHS clusters in Africa, colored by NTL.** Base map data from the spData package in R, derived from Natural Earth (public domain). Nighttime lights data (NTL) is derived from Li et al. (2020). The figure depicts geolocated DHS clusters across Africa, overlaid with harmonized NTL data. Clusters are colored by NTL intensity (low to high), with darker shades representing higher light emissions. Countries included in the study are shaded in gray. The figure was created by the author.

We aggregate these rasters at a coarser resolution of 10 km$^2$ to match them with DHS household wealth data using the geographic coordinates at the cluster level in the DHS surveys. To provide additional context, we overlay country boundaries on the NTL data in Fig 2. This overlay allows us to visualize the within-country variation in NTL and compare it with the within-country variation of the Wealth Index as well as HDI and GDP per capita.

**Population density data.** Population density data were sourced from the CIESIN Gridded Population of the World (GPWv4) [32] at a spatial resolution of 5 km, approximately 2.5 arc minutes, covering intervals of 5 years. For the years between these intervals, we implemented linear interpolation to estimate population density following the methods from previous research [37]. Similar to NTL, these were then aggregated to 10 km$^2$ and combined with DHS data for the years 2004 to 2019.

**Alternative dependent variables: HDI and GDP per capita.** Along with the DHS wealth index, we use two alternative dependent variables: GDP per capita and the HDI. GDP per capita stands as a core metric for gauging economic performance and is widely employed as an indicator of average living standards or economic prosperity. HDI is often utilized to categorize countries based on their human development levels (health, educational attainment and standard of living) [38], and is a more holistic measure compared to just income or wealth alone. However, the annual release of official global HDI estimates by the Human Development Report Office of the United Nations Development Programme (UNDP) is limited to highly aggregated national level estimates, hindering its application in scenarios requiring

sub-national details. Despite being considered a more meaningful metric than income alone, HDI has not replaced income measures for assessing development progress within countries due to lack of availability at sub-national levels. Additionally, the reliance on slow, infrequent, and costly global-scale ground-based data collection for all current HDI estimates severely limits their practical usability beyond cross-national rankings [39].

Recognizing this limitation, subnational annual gridded datasets for GDP per capita and HDI have been produced for the whole world at a spatial resolution of 5 arc-min level (10 km) [7]. For the GDP per capita, these datasets combine both sub-national (based on [40]) and national datasets (from the World Bank dataset and CIA's World Factbook). Priority is given to reported sub-national data, followed by the utilization of interpolated and extrapolated sub-national data, in conjunction with national averages. For HDI, scaling factors are devised to integrate information from both sub-national and national sources, using national-level data (from UNDP) and subnational-level (from UNDP and census reports where available).

We extracted both these datasets (from [7]) for the years for which they were available for our sample (2004 to 2015). Below, we plot two maps showing the variation in GDP per capita (Fig 3) and HDI data (Fig 4) for the DHS cluster locations. We divide the data into 5 bins of equal intervals (similar to 5 bins of the wealth index) to plot them. Note that there is very little variation within countries for both datasets. This goes against the motivation of creating these datasets to explain sub-national variation.

**Descriptive statistics.** Table 2 presents summary statistics for the full sample in panel (a), for the urban sample in panel (b), and for the rural sample in panel (c). On average, urban

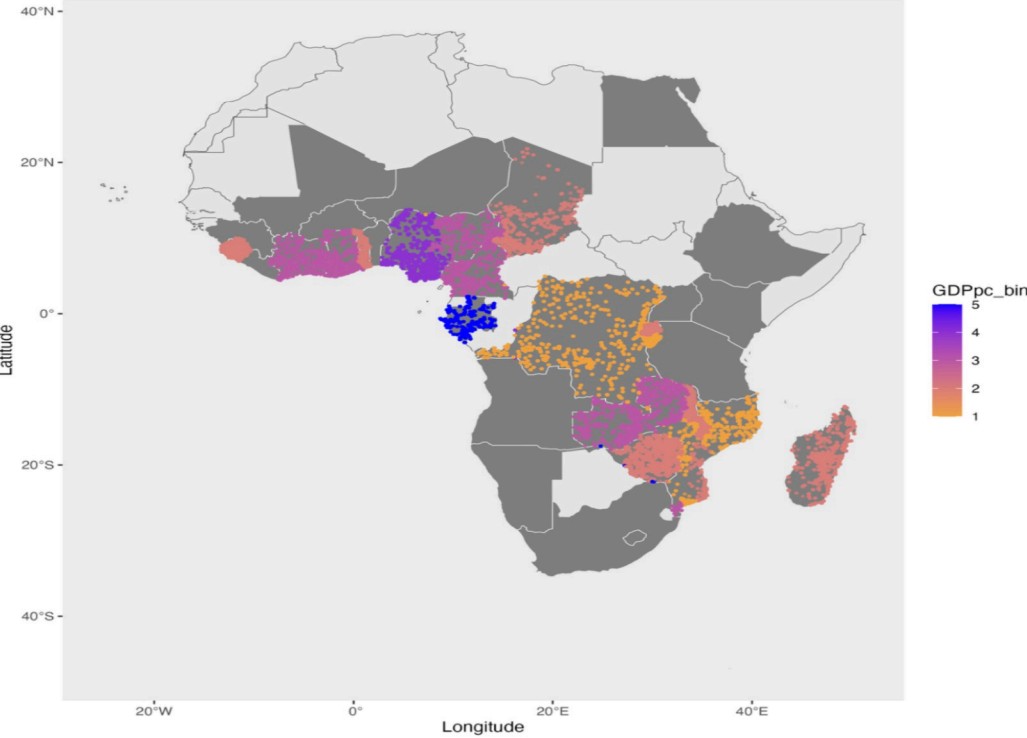

**Fig 3. Geolocated DHS clusters in Africa, colored by GDP per capita.** Base map data from the spData package in R, derived from Natural Earth (public domain). Gross Domestic Product per capita (GDPpc) data is from Kammu et al. (2018). The figure depicts geolocated DHS clusters across Africa, categorized by GDPpc quintiles. Darker colors indicate higher income levels. Countries shaded in gray represent those included in the study. The figure was created by the author.

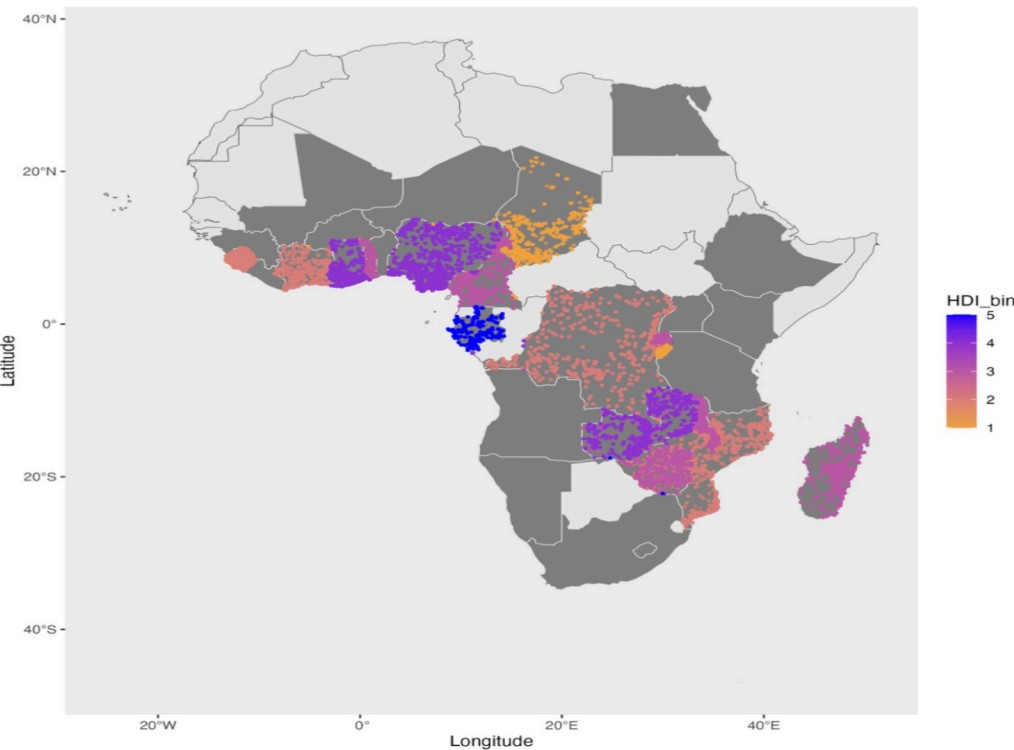

**Fig 4. Geolocated DHS clusters in Africa, colored by HDI.** Base map data from the spData package in R, derived from Natural Earth (public domain). Human Development Index (HDI) is sourced from Kammu et al. (2018). The figure depicts geolocated DHS clusters in Africa, categorized by HDI quintiles. Darker colors indicate higher levels of human development. Countries shaded in gray represent those included in the analysis. The figure was created by the author.

areas emit approximately five times more light (23) than rural areas (5.7). The standard deviation of 13 indicates that there is a lower degree of variability in the level of NTL across different rural areas compared to 21 in urban areas. Similarly, urban areas are, on average, over nine times more densely populated (3,250) than rural areas (363). The standard deviation of 984 in rural areas is also smaller than that in urban areas.

In Fig 5, we present the Spearman correlation coefficient matrix for our variables. As anticipated, there is a strong positive correlation between NTL and population density (0.7). Similarly, we find a high correlation between the HDI and GDP per capita (0.8). When examining the wealth index, NTL display a stronger correlation (0.7) compared to population density (0.5). Additionally, both HDI and GDP per capita show a positive correlation of 0.5 with the wealth index. Notably, while NTL exhibit a relatively high correlation (0.5) with HDI and GDP per capita, population density shows a lower correlation of 0.2 with both indicators.

## Methods

Our objective is to evaluate the extent to which harmonized NTL data serve as a proxy for socio-economic outcomes at small spatial levels. To achieve this objective, we employ repeated $k$-fold cross-validation to assess the predictive accuracy of our models. This method involves randomly splitting the data into **k = 10** equally sized folds, and the process is repeated **10 times**.

The models are trained on $k$-1 folds and tested on the remaining fold for each iteration of the cross-validation process. Predictions are then made on the test data, and performance

**Table 2. Summary statistics.**

(a) Full Sample

| Variable | Obs. | Mean | Std. Dev. | Min. | Max. | Median |
|---|---|---|---|---|---|---|
| Mean Wealth Index | 39072 | 3 | 1.2 | 1 | 5 | 2.8 |
| Nightlights | 39072 | 12 | 18 | 0 | 63 | 2.7 |
| Population Density | 39072 | 1418 | 4262 | 0 | 45196 | 195 |
| Human Development Index | 30880 | 0.5 | 0.095 | 0.29 | 0.7 | 0.49 |
| GDP per capita | 30880 | 3551 | 3418 | 365 | 17595 | 1876 |

(b) Urban Sample

| Variable | Obs. | Mean | Std. Dev. | Min | Max | Median |
|---|---|---|---|---|---|---|
| Mean Wealth Index | 14278 | 4.1 | 0.86 | 1 | 5 | 4.4 |
| Nightlights | 14278 | 23 | 21 | 0 | 63 | 14 |
| Population Density | 14278 | 3250 | 6538 | 0 | 45196 | 692 |
| Human Development Index | 11160 | 0.51 | 0.1 | 0.31 | 0.69 | 0.5 |
| GDP per capita | 11160 | 4346 | 3769 | 365 | 17595 | 2614 |

(c) Rural Sample

| Variable | Obs. | Mean | Std. Dev. | Min | Max | Median |
|---|---|---|---|---|---|---|
| Mean Wealth Index | 24794 | 2.4 | 0.81 | 1 | 5 | 2.3 |
| Nightlights | 24794 | 5.7 | 13 | 0 | 63 | 0 |
| Population Density | 24794 | 363 | 984 | 0 | 42424 | 109 |
| Human Development Index | 19720 | 0.49 | 0.091 | 0.29 | 0.7 | 0.48 |
| GDP per capita | 19720 | 3101 | 3113 | 365 | 17595 | 1649 |

Notes: NTL are unitless, measured in Digital Number ranging from 0 to 63. These values represent the intensity of the lights, with higher values indicating greater intensity. Population density is measured as the number of people per square kilometer. Wealth Index is an index between 1 to 5. HDI is an index between 0 and 1, and GDP per capita is expressed in constant 2011 US dollars (constant 2011). Household Wealth Index data is derived from DHS. Nighttime lights data is derived from Li et al. (2020). Population density is sourced from the GPWv4, Human Development Index (HDI) and Gross Domestic Product per capita data are both from Kummu et al. (2018). For more details, see Table 1.

metrics, such as the R-squared, are computed. This iterative process ensures that each fold is used exactly once as the test dataset. Finally, the out-of-sample R-squared is calculated as the average performance metric across all iterations of the cross-validation process. This metric represents the proportion of the variation in the response variable that is explained by the model when applied to unseen data, thereby serving as an estimate of the model's predictive accuracy on new observations.

We begin by estimating a relationship using Ordinary Least Squares regression, which enables us to examine key associations, including the relationship between NTL with socio-economic outcomes for the pooled sample. Socio-economic outcomes are defined using the three dependent variables measured in this paper: the DHS wealth index, HDI, and GDP per capita. We estimate the models with and without controlling for population density to evaluate the contribution of NTL independently of population density.

Additionally, we examine these relationships within two distinct subsamples: urban and rural. However, we only present results for urban versus rural subsamples in the case of the DHS wealth index, as we found very little difference in the variation captured by NTL for HDI and GDP per capita between urban and rural areas.

Furthermore, to account for the unique characteristics specific to each country and year over the study period, we incorporate country and time fixed effects in all regressions, particularly in those where the DHS wealth index serves as the dependent variable. This is essential

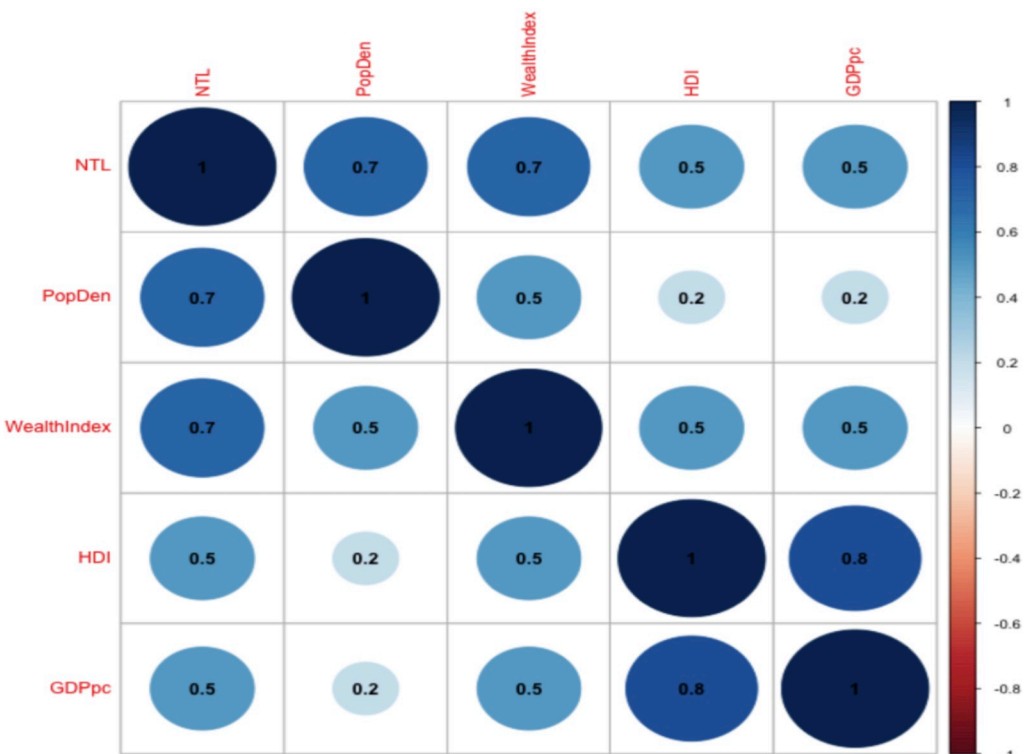

**Fig 5. Correlation plot.** Author's calculations for pairwise correlations among key variables. Wealth Index, Nightime Light (NTL), Population Density (PopDen), Human Development Index (HDI), and Gross Domestic Product per capita (GDPpc). Wealth Index data is derived from DHS. NTL data is derived from Li et al. (2020). PopDen is sourced from the GPWv4, HDI and GDPpc data are both from Kummu et al. (2018). For more details, see Table 1. The size and color intensity of the circles indicate the strength and direction of the correlations, with darker blue representing stronger positive correlations and lighter shades or red (if present) indicating weaker or negative correlations.

due to the lack of direct comparability of the wealth index across different countries or even repeated rounds of survey for the same country. For example, DHS surveys often employ different methodologies over time, making results from different survey rounds potentially inconsistent. For a detailed list of the survey rounds used for each country, refer to S1 Table.

We estimate the following relationship:

$$Y_{cjt} = \beta_0 + \beta_1 NTL_{cjt} + \beta_2 \ populationdensity_{cjt} + \Gamma_j + \Gamma_t + e_{cjt} \qquad (1)$$

where $Y_{cjt}$ represents the socio-economic outcomes in cluster $c$, country $j$, and year $t$, which is measured by three dependent variables—the DHS wealth index, HDI, and GDP per capita. The right hand side variables include: $NTL_{cjt}$ which is night light intensity; $populationdensity_{cjt}$ which is population density; $\Gamma_j$ and $\Gamma_t$ are country and year fixed effects, respectively, and $e_{cjt}$ is the error term. Adding country and year fixed effects allows us to explore the relationship of NTL with socio-economic outcomes within specific country and year groups, effectively examining variations within a single survey. Standard errors are clustered at the same level of fixed effects (country and year) which will adjust for potential correlations between error terms within the same country or year.

In the main analysis, all regressions with HDI and GDP per capita as the dependent variables include both country and year fixed effects. However, for robustness, we also present

results in the (S2–S7 Tables) for models estimated under alternative specifications: (i) without fixed effects; (ii) with only country fixed effects; and (iii) with only year fixed effects.

We observe that the NTL, population density, and GDP per capita data exhibit right-skewed distributions. To address this skewness, we initially consider using the logarithm of their values as a smoothing technique. However, due to a considerable proportion of zero observations, dropping these observations from our samples is not feasible. To address this challenge while still accommodating zero-valued observations, we opt for the Inverse Hyperbolic Sine (IHS) transformation. This preserves the properties of the log transformation while allowing us to retain zero-valued observations [41].

## Results

Table 3 displays the outcomes of regressing the DHS wealth index on harmonized NTL, both without and with population density (Columns 1 and 3 respectively). Additionally, Column (2) shows the results of regressing the DHS wealth index only on population density. Since NTL and population density have been transformed using the IHS transformation, we can interpret them as percentage changes.

The findings in Table 3 reveal that variations in NTL alone significantly explain variations in the wealth index, yielding an Out-of-Sample R-Squared (OOS-R Squared) of 48% (Column 1). When population density is included as a control variable (Column 3), the OOS-R Squared increases slightly to 49%, suggesting that models incorporating both NTL and population density marginally outperform those without population density.

It is important to note that the relationships estimated do not imply causation; the findings only indicate that higher levels of NTL are associated with higher wealth, holding all other factors constant. All specifications in the analysis include country and year fixed effects, allowing for within-country comparisons. This is essential because the wealth index is not comparable across countries.

**Table 3. OLS regression results with country and year fixed effects.**

|  | Wealth Index | | |
|---|---|---|---|
|  | **(1)** | **(2)** | **(3)** |
| Nightlights | 0.573*** |  | 0.475*** |
|  | (0.002) |  | (0.005) |
| Population Density |  | 0.385*** | 0.104*** |
|  |  | (0.003) | (0.005) |
| Fixed effects (Country and year) | Yes | Yes | Yes |
| OOS $R^2$ | 0.480 | 0.352 | 0.491 |
| Adjusted $R^2$ | 0.479 | 0.351 | 0.490 |
| Residual Std. Error | 0.839 | 0.937 | 0.830 |

Notes: Dependent variable is the DHS mean wealth index. Nightlights and population density have been transformed using the inverse hyperbolic sine (IHS) transformation. Standard errors (in parentheses) are clustered at the country and year level. All models have country and year fixed effects.

*$p<0.1$;

**$p<0.05$;

***$p<0.01$.

Household Wealth Index data is derived from DHS. Nighttime lights data is derived from Li et al. (2020). Population density is sourced from the GPWv4. For more details, see Table 1.

To address potential concerns about spatial autocorrelation in the residuals, we conducted supplementary analyses using spatial fixed effects models. Due to computational constraints, spatial models were estimated on a subset of the data. Importantly, the results from these spatial models (see S8 Table) were consistent with the primary analysis presented in Table 3, further reinforcing the robustness of our findings.

It is crucial to conduct separate analyses for urban and rural areas due to differences in their economic activity types, their population density, and lighting characteristics. Previous research [30] observed that satellite-detected NTL mainly represent urban economic activity, consisting of concentrated street lamps and industrial facilities typical of urban settings, while such lights are rarely found in rural villages. To address this, we divided our sample into urban and rural regions and presented the results separately in Table 4. The table includes the effects of NTL, population density, and their combined influence on wealth in urban versus rural areas. While previous research [30] only used VIIRS data for two years and focused exclusively on Indonesia, our analysis utilizes the harmonized NTL dataset (DMSP + VIIRS) for 34 countries, spanning 2004 to 2019. This harmonized dataset enables us to study long-term changes in nighttime light intensity, addressing inconsistencies between the two sources.

Our findings (from Table 4) show that the share of variation in the wealth index explained by NTL in urban areas (43%) (Column 3) is over two times that in rural areas (21%) (Column 6). Similarly, the explanatory power in models with population density is over two times for urban areas (36%) (Column 4) as compared to rural (16%) (Column 7). Furthermore, examining the direction and relative strength of correlations reveals that both NTL and population density exhibit positive correlations with the wealth index, with the NTL consistently demonstrating a significantly stronger correlation across all models.

In both urban and rural settings, the inclusion of population density as a control variable leads to a reduction in the magnitude of the coefficient on NTL but it remains significant. This observation is likely attributable to the fact that although part of the variation in NTL is absorbed by variation in population density within the DHS clusters, NTL still retains

**Table 4. OLS regression for wealth index in urban v/s rural areas.**

| | Mean Wealth Index | | | | | | | |
|---|---|---|---|---|---|---|---|---|
| | Overall | | Urban | | | Rural | | |
| | (1) | (2) | (3) | (4) | (5) | (6) | (7) | (8) |
| Nightlights | 0.572*** | | 0.317*** | | 0.263*** | 0.352*** | | 0.280*** |
| | (0.003) | | (0.005) | | (0.009) | (0.007) | | (0.008) |
| Population Density | | 0.384*** | | 0.181*** | 0.049*** | | 0.192*** | 0.096*** |
| | | (0.004) | | (0.004) | (0.006) | | (0.006) | (0.006) |
| Fixed effects (Country and year) | Yes | Yes | Yes | Yes | Yes | Yes | Yes | Yes |
| OOS R$^2$ | 0.481 | 0.352 | 0.433 | 0.365 | 0.439 | 0.212 | 0.171 | 0.226 |
| Adjusted R$^2$ | 0.480 | 0.351 | 0.430 | 0.362 | 0.436 | 0.210 | 0.168 | 0.224 |
| Residual Std. Error | 0.856 | 0.936 | 0.649 | 0.686 | 0.646 | 0.719 | 0.729 | 0.707 |

Notes: Dependent variable is the DHS mean wealth index. Nightlights and population density have been transformed using the inverse hyperbolic sine (IHS) transformation. Standard errors (in parentheses) are clustered at the country and year level. All models have country and year fixed effects.

*p<0.1;

**p<0.05;

***p<0.01.

Household Wealth Index data is derived from DHS. Nighttime lights data is derived from Li et al. (2020). Population density is sourced from the GPWv4. For more details, see Table 1.

substantial information about the wealth index that surpasses the influence of population density. The estimates presented in Table 4 are consistent with the notion that lights serve as a useful proxy for urban economic activity. Additionally, variation in lights explains a modest proportion (21%) of the variation in the wealth index in rural areas. Overall, our findings using the harmonized NTL data are broadly consistent with previous research (see [30]) for both urban and rural areas. For instance, in their study for Indonesia [30], they found that for the urban sector, the relationship between NTL and economic activity remains positive regardless of the type of NTL data used. However, in rural areas, economic activity was negatively related to DMSP NTL, while it was positively but imprecisely related to VIIRS NTL. They reported very small R-squared values for rural areas—3% using DMSP and 1% using VIIRS—with the results for VIIRS being statistically insignificant. In contrast, they found much higher R-squared values for urban areas—36% using DMSP and 68% using VIIRS—suggesting that NTL is a better proxy for urban than rural economic activity.

Our results outperform theirs for both DMSP and VIIRS, likely due to the use of harmonized NTL data, which addresses inconsistencies between the two sources. However, it is important to note that the dependent variable in earlier research [30] is regional GDP, whereas our analysis focuses on the DHS wealth index. Thus, the results are not directly comparable.

Next, we estimated Eq 1 using the GDP per capita and Human Development Index (HDI) as the dependent variables in Tables 5 and 6 respectively. In Table 5, we employ the new Harmonized Nighttime Lights (NTL) dataset alongside population density to analyze the variation in GDP per capita within a specific country and year group, incorporating country and year fixed effects.

Given the transformation of NTL, population density, and GDP per capita using the Inverse Hyperbolic Sine (IHS) transformation, the coefficient estimates are interpreted as elasticities. The estimation of the model in column 1, yields a precisely estimated elasticity of NTL of 0.035, accompanied by a high OOS R-square of 93 percent. However, it is important to note that these results should not be interpreted as causal. Instead, they suggest that a 1% increase

**Table 5. OLS regression for GDP per capita with country and year fixed effects.**

|  | Gross Domestic Product per capita | | |
|---|---|---|---|
|  | **(1)** | **(2)** | **(3)** |
| Nightlights | 0.035*** |  | 0.036*** |
|  | (0.001) |  | (0.002) |
| Population Density |  | 0.020*** | -0.002 |
|  |  | (0.001) | (0.001) |
| Fixed effects (Country and year) | Yes | Yes | Yes |
| OOS R$^2$ | 0.935 | 0.933 | 0.935 |
| Adjusted R$^2$ | 0.935 | 0.934 | 0.935 |
| Residual Std. Error | 0.220 | 0.222 | 0.220 |

Notes: Dependent variable is the Gross Domestic Product per capita. Gross Domestic Product per capita, Nightlights, and population density have been transformed using the inverse hyperbolic sine (IHS) transformation. Standard errors (in parentheses) are clustered at the country and year level. All models have country and year fixed effects.

*p<0.1;

**p<0.05;

***p<0.01.

Nighttime lights data is derived from Li et al. (2020). Population density is sourced from the GPWv4, and Gross Domestic Product per capita data from Kummu et al. (2018). For more details, see Table 1.

**Table 6. OLS regression for HDI with country and year fixed effects.**

| | Human Development Index | | |
|---|---|---|---|
| | (1) | (2) | (3) |
| Nightlights | 0.0002*** | | 0.001*** |
| | (0.0001) | | (0.0001) |
| Population Density | | -0.0002* | -0.0004*** |
| | | (0.00004) | (0.0001) |
| Fixed effects (Country and year) | Yes | Yes | Yes |
| OOS R$^2$ | 0.984 | 0.984 | 0.985 |
| Adjusted R$^2$ | 0.984 | 0.984 | 0.984 |
| Residual Std. Error | 0.011 | 0.011 | 0.011 |

Notes: Dependent variable is the Human Development Index. Nightlights and population density have been transformed using the inverse hyperbolic sine (IHS) transformation. Standard errors (in parentheses) are clustered at the country and year level. All models have country and year fixed effects.

*p<0.1;

**p<0.05;

***p<0.01.

Nighttime lights data is derived from Li et al. (2020). Population density is sourced from the GPWv4, and Human Development Index (HDI) from Kummu et al. (2018). For more details, see Table 1.

in NTL is associated with a 3.4% higher GDP per capita, all else being equal. Similarly, a 1% increase in population density is associated with a 2% higher GDP per capita.

The observed variation in GDP per capita is primarily driven by between-country differences rather than within-country disparities. Specifically, controlling for baseline differences between countries through country fixed effects alone explains 91% of the observed variation, as detailed in S2 Table. Furthermore, the consistent OOS R-square across models, regardless of the inclusion of population density, indicates that NTL possesses significant predictive power on its own, independent of population density. In fact, when both NTL and population density are included in the regression (Column 3), population density is not statistically significant and even exhibits a sign inconsistent with expectations.

We use an alternative indicator of sub-national economic development. The HDI is often considered a more meaningful metric than income or wealth alone. However, there are few studies [28, 29] that have tried to explain how well NTLs predict HDI. Using the new sub-national global HDI dataset ([7]), we study the variation in HDI explained by NTL and population density within a specific country and year group, employing country and year fixed effects (Table 6).

The model estimated in column 1 explains a substantial proportion of the variation in HDI, with a high OOS R-square of 98%. This suggests that nightlights serve as a suitable proxy for HDI, given significant correlations across all specifications. As with GDP per capita in Table 5, after including population density in the regression (column 3), NTL continues being positively and significantly correlated with HDI, while population density exhibits a wrong sign (and now is statistically significant).

A previous study [28] examined potential channels through which a positive association between NTL and HDI may have taken effect, highlighting improved schooling outcomes, lower infant mortality rates, and increased local economic activity in areas with higher NTL. In our models, the inclusion of only country fixed effects accounts for up to 92% of the variation in HDI, as detailed in S3 Table. This highlights that the minimal within-country variation in HDI is primarily driven by geographical heterogeneity. Between-country differences,

rather than within-country disparities, explain a substantial portion of the observed variation in HDI.

## Conclusion

Access to consistent and accurate data over long time periods is crucial for understanding trends and formulating informed policy decisions at regional or local levels. For example, analyzing local economic growth patterns or assessing the effects of climate change on regional ecosystems necessitates spatially disaggregated, historical data. NTL data can serve as valuable proxies for measuring development at local levels where alternative indicators may be lacking. The DMSP (discontinued in 2013) and the VIIRS are two widely utilized series of NTL data. While VIIRS data offer improvements over DMSP, including higher precision and updated technology, DMSP is still widely used in economics literature due to its longer time series spanning from 1992 to 2013. Newly available harmonized NTL data are therefore essential to bridging the gap between DMSP and VIIRS datasets.

We utilized a harmonized dataset [25], which integrates DMSP and VIIRS data to offer a continuous NTL time series from 1992 to 2021. This dataset, characterized by its relatively new processing technology and higher precision [42], provides a consistent framework for analyzing economic trends with high spatial resolution. To the best of our knowledge, our research is the first to test the accuracy of the harmonized NTL dataset in measuring economic activity using household wealth index from the DHS, and alternatively through other indicators of subnational economic development; namely, gridded HDI and GDP per-capita.

Through our analysis, we make four key contributions to the literature. First, we test the accuracy of the new harmonized NTL dataset in measuring economic activity at a small spatial scale in developing country municipalities. Second, we assess the extent to which NTL vary independently of population density. Third, we explore whether NTL can serve as a proxy for human development (HDI) beyond wealth or income, an area largely unexplored in previous research. Lastly, our study adds to the ongoing debate on the utility of NTL as a proxy for economic activity at subnational levels, specifically focusing on urban versus rural areas.

In evaluating predictive performance of our models on unseen data (new observations), we use $k$-fold repeated cross validation to calculate out-of-sample R-squared. The results demonstrate that the harmonized NTL data can serve as valuable indicators for studying wealth index at local levels, with strong explanatory power. Moreover, models that control for population density slightly increase the variation explained by NTL in wealth index. Our analysis of the relationship between NTL and wealth index across urban and rural settings reveal that the share of variation explained by NTL in wealth index is two times higher in urban areas as compared to rural areas. This finding is in line with the literature that finds NTL to be a reliable proxy only for urban areas. Furthermore, using a unique global gridded dataset for both GDP per capita and HDI, our study demonstrates that NTL serve as reliable proxies for both GDP per capita and the HDI at subnational levels. Importantly, both estimated coefficients and predictive power of NTL remains significant even after controlling for population density. Additionally, we note that the observed high variation explained in our models explaining both GDP per capita and HDI is primarily driven by between-country differences rather than within-country differences.

The implications of this research extend beyond the field of economics to various disciplines requiring sub-national data on economic prosperity. For instance, in the context of climate change, extreme weather events have highly localized impacts that are often obscured when using aggregated GDP data. Using the new harmonized NTL data, which allows studying long time-periods and captures localized economic activity, can help researchers gain a

better understanding of the effects of climate change at sub-national levels. Overall, the insights provided in this study can guide researchers in selecting appropriate indicators for their specific contexts and research objectives. This, in turn, contributes to more informed decision-making and policy formulation.

## Supporting information

**S1 Appendix. Harmonized nightlights data.**
(DOCX)

**S2 Appendix. Literature review.**
(DOCX)

**S1 Table. List of countries and DHS waves in the sample.**
(DOCX)

**S2 Table. OLS regression for GDP per capita using country fixed effects.**
(DOCX)

**S3 Table. OLS regression for HDI using country fixed effects.**
(DOCX)

**S4 Table. OLS regression for GDP per capita with year fixed effects.**
(DOCX)

**S5 Table. OLS regression for HDI with year fixed effects.**
(DOCX)

**S6 Table. OLS regression for GDP.** No fixed effects.
(DOCX)

**S7 Table. OLS regression for HDI.** No fixed effects.
(DOCX)

**S8 Table. Results from the spatial model with country and year fixed effects.**
(DOCX)

## Acknowledgments

The authors sincerely thank Dr. Shanjukta Nath for her guidance in selecting the methodology for this study. We are also deeply grateful to Dr. Nicole Gottdenker for her invaluable comments and constructive feedback, which greatly improved this work. Additionally, we would like to extend our heartfelt thanks to Dr. Aditi Kadam for her generous assistance during the initial phases of this project.

## Author Contributions

**Conceptualization:** Susana Ferreira, Patrick Stephens, Mekala Sundaram, Jonathan Wilson.

**Data curation:** Prachi Jhamb.

**Formal analysis:** Prachi Jhamb.

**Funding acquisition:** Susana Ferreira, Patrick Stephens, Mekala Sundaram.

**Investigation:** Prachi Jhamb.

**Methodology:** Prachi Jhamb, Susana Ferreira, Patrick Stephens, Mekala Sundaram.

**Project administration:** Prachi Jhamb, Susana Ferreira, Patrick Stephens, Mekala Sundaram.

**Resources:** Prachi Jhamb, Susana Ferreira, Patrick Stephens, Mekala Sundaram, Jonathan Wilson.

**Software:** Prachi Jhamb, Jonathan Wilson.

**Supervision:** Susana Ferreira.

**Validation:** Prachi Jhamb.

**Visualization:** Prachi Jhamb.

**Writing – original draft:** Prachi Jhamb.

**Writing – review & editing:** Prachi Jhamb, Susana Ferreira, Patrick Stephens, Mekala Sundaram.

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
