## [Decision Letter · Decision Letter 0]

10 Oct 2024

PONE-D-24-17551Shedding Light on Development: Leveraging the new Nightlights data to measure Economic Progress?PLOS ONE

Dear Dr. Jhamb,

Thank you for submitting your manuscript to PLOS ONE. After careful consideration, we feel that it has merit but does not fully meet PLOS ONE’s publication criteria as it currently stands. Therefore, we invite you to submit a revised version of the manuscript that addresses the points raised during the review process.

**The paper has received positive feedback from reviewers as a quality paper meeting the PLOS one criteria for relevance of the topic and scientific rigor. However, one of the reviewers suggest to check for spatial autocorrelation that you might consider or comment about.      **

We look forward to receiving your revised manuscript.

Kind regards,

Martin Ramirez-Urquidy, PhD. Economics

Academic Editor

PLOS ONE

**Journal Requirements:**

This  work  was  supported  by  National Institutes of Health, NIH  R01Al156866  ‘Spillover  of  Ebola  and  other  filoviruses  at  ecological  boundaries’  (Patrick Stephens lead investigator).” https://www.nih.gov/ 

The funder played no role in the study design, data collection or analysis, decision to publish, or preparation of the manuscript.

4. We note that Figures 1, 2, 3 and 4 in your submission contain map images which may be copyrighted. All PLOS content is published under the Creative Commons Attribution License (CC BY 4.0), which means that the manuscript, images, and Supporting Information files will be freely available online, and any third party is permitted to access, download, copy, distribute, and use these materials in any way, even commercially, with proper attribution. For these reasons, we cannot publish previously copyrighted maps or satellite images created using proprietary data, such as Google software (Google Maps, Street View, and Earth). For more information, see our copyright guidelines: http://journals.plos.org/plosone/s/licenses-and-copyright.

We require you to either present written permission from the copyright holder to publish these figures specifically under the CC BY 4.0 license, or remove the figures from your submission:

a. You may seek permission from the original copyright holder of Figures 1, 2, 3 and 4 to publish the content specifically under the CC BY 4.0 license.  

5. Please include your tables as part of your main manuscript and remove the individual files. Please note that supplementary tables (should remain/ be uploaded) as separate "supporting information" files

**Additional Editor Comments:**

The paper has received positive feedback from reviewers as a high-quality paper meeting the PLOS one criteria of relevance of the topic and scientific rigor. However, one of the reviewers suggest to check for spatial autocorrelation that you might consider.

Reviewers' comments:

Reviewer's Responses to Questions

**Comments to the Author**

1. Is the manuscript technically sound, and do the data support the conclusions?

Reviewer #1: Yes

Reviewer #2: Yes

2. Has the statistical analysis been performed appropriately and rigorously? 

Reviewer #1: No

Reviewer #2: Yes

3. Have the authors made all data underlying the findings in their manuscript fully available?

Reviewer #1: Yes

Reviewer #2: Yes

4. Is the manuscript presented in an intelligible fashion and written in standard English?

Reviewer #1: Yes

Reviewer #2: Yes

5. Review Comments to the Author

**Reviewer #1:** This paper is interesting and contains pertinent information, but it is necessary to review the econometric model because the georeferenced information from neighboring cases can generate the problem of spatial autocorrelation, which limits the validity of the regression coefficients. There is a very rich methodology on econometric models with spatial data. You can consult:

Baltagi, B.H., Li, D. (2004). Prediction in the Panel Data Model with Spatial Correlation. In: Anselin, L., Florax, R.J.G.M., Rey, S.J. (eds) Advances in Spatial Econometrics. Advances in Spatial Science. Springer, Berlin, Heidelberg. https://doi.org/10.1007/978-3-662-05617-2_13

Anselin, L., Gallo, J.L., Jayet, H. (2008). Spatial Panel Econometrics. In: Mátyás, L., Sevestre, P. (eds) The Econometrics of Panel Data. Advanced Studies in Theoretical and Applied Econometrics, vol 46. Springer, Berlin, Heidelberg. https://doi.org/10.1007/978-3-540-75892-1_19

**Reviewer #2:** The document raises a topic that is not innovative: the correlation between light intensity, economic activity and economic growth that translates into economic development. Recently, for example, the institutionalist vision of authors such as Acemoglu & Robinson grants special importance to regulatory frameworks, the development of infrastructure and the formation of quality of life indicators.

Notwithstanding the above, the document has important strengths, which are listed below: 1) a robust, adequate and updated literature review; 2) a clear and consistent wording with the design of a justified and relevant research problem; 3) an interesting delimitation of the object of study, being interested in the countries of sub-Saharan Africa; 4) a robust and innovative methodological proposal with a consistent estimation process; 5) obtaining and analyzing results with a purposeful perspective and contribution to knowledge.

The study highlights the relevant evidence regarding the unnecessary correlation of urban lighting presence with population density. Such a result seems to suggest a development dynamic disconnected from quality of life indicators, especially in communities with greater social backwardness in Africa, whose implications may be extensive to different regions of the underdeveloped world.

It is, in summary, a document with all the possibilities of publication, which contributes to the literature, with a well-applied methodology and from which economic policy implications are derived that motivate a reflection on the nature, type and scope of development in regions, which are referents of exclusion in the great processes imposed by globalization.

Finally, although the tabular and graphic schemes are of our own creation, it is suggested that the reader be told the source of the data in order to specify the origin of the information contained in Table 1 in each table and in each graph.

6. PLOS authors have the option to publish the peer review history of their article (what does this mean?). If published, this will include your full peer review and any attached files.

Reviewer #1: No

Reviewer #2: **Yes: **Edgar David Gaytan-Alfaro

---

## [Author Response · Author response to Decision Letter 0]

1 Jan 2025

Responses to Reviewer #1’s comments: Thank you for highlighting this important concern. To address this concern, we conducted Moran’s I tests and Monte Carlo simulations on the residuals of all models in our analysis to determine the presence and significance of spatial autocorrelation. Below are the key findings: Moran’s I Tests on Residuals:

The Moran’s I values for the three models (from Table 3 in main text) are approximately 0.422, 0.501, and 0.422, respectively, with p-values < 2.2e-16. These results indicate a moderate but statistically significant level of positive spatial autocorrelation in the residuals of the three models. 

 To address this issue, we re-estimated the model using spatial fixed effect models (“spaMM” package in R). Before diving in, we would like to provide some context and clarify our methodological choices. Initially, we aimed to incorporate spatial fixed effect models across the full dataset to account for spatial autocorrelation. Unfortunately, due to the size of the dataset and computational limitations, running these models was not feasible; the process required extensive computational time (up to 72 hours per model) and ultimately resulted in crashing of the software. To address this, we conducted supplementary analyses using spatial econometric models on a subset of the dataset (15,000 observations). This allowed us to evaluate spatial autocorrelation in a more manageable framework. The results of these supplementary spatial econometric models (Table R.3) are consistent with the primary analysis (Table 3 in the main text) and serve as a robustness check. The results of these supplementary spatial econometric models are presented in the supporting information document as S3 Table 8.

Our findings are broadly consistent with those of Baltagi et al. (2004), who demonstrated that accounting for spatial autocorrelation does not always improve prediction accuracy. In their analysis of liquor demand, the fixed effect model without spatial autocorrelation achieved the lowest RMSE (0.1360), closely followed by the random effect model without spatial autocorrelation (0.1367). For short-term forecasts (1 year ahead), incorporating spatial autocorrelation improved performance marginally. However, for longer-term forecasts (2 or more years), the FE and RE estimators without spatial autocorrelation consistently outperformed models that accounted for spatial auto-correlation. Overall, Baltagi et al. (2004) concluded that taking into account heterogeneity across states using FE or RE estimators yielded the best out-of-sample RMSE forecast performance. While spatial autocorrelation provided marginal improvements in the first year, its predictive benefits were minimal beyond that, and the differences between spatial and non-spatial models were not statistically significant.

We have included the results of the supplementary spatial econometric models (S3 Table 8) for the subset of the data to provide additional context and support for our findings. However, the primary models in the manuscript rely on fixed effects without accounting for spatial autocorrelation. We appreciate your understanding of these methodological challenges and believe that the inclusion of supplementary spatial analyses strengthens the validity of our approach. We are happy to provide further details or clarifications if needed.

Responses to Reviewer #2’s comments: Thank you for highlighting the need for clearer attribution of data sources in tables and graphs. While Table 1 presents the source of each variable used in our study, we have also revised all tables and figures to include specific data source details and ensure clarity for the audience.

Once again, we appreciate your constructive comments, which have helped us improve the manuscript. We hope the revisions will address your concerns.

---

## [Editor Report · Decision Letter 1]

17 Jan 2025

Shedding light on development: Leveraging the new nightlights data to measure economic progress

PONE-D-24-17551R1

Dear Dr. Jhamb,

We’re pleased to inform you that your manuscript has been judged scientifically suitable for publication and will be formally accepted for publication once it meets all outstanding technical requirements.

Kind regards,

Martin Ramirez-Urquidy, PhD. Economics

Academic Editor

PLOS ONE
---

## [Editor Report · Acceptance letter]

23 Jan 2025

PONE-D-24-17551R1 

PLOS ONE

Dear Dr. Jhamb, 

I'm pleased to inform you that your manuscript has been deemed suitable for publication in PLOS ONE. Congratulations! Your manuscript is now being handed over to our production team.

Kind regards, 

on behalf of

Dr. Martin Ramirez-Urquidy 

Academic Editor

PLOS ONE